

# Clinical streptococcal isolates, distinct from *Streptococcus pneumoniae,* but containing the β-glucosyltransferase *tts* gene and expressing serotype 37 capsular polysaccharide

Carmen L. Sheppard[1], Georgia Kapatai[1], Karen Broughton[1], Ulf Schaefer[2], Matthew Hannah[3], David J. Litt[1] and Norman K. Fry[1]

[1] Respiratory and Vaccine Preventable Bacteria Reference Unit, Public Health England, London, United Kingdom
[2] Infectious Disease Informatics/Bioinformatics, Public Health England, London, United Kingdom
[3] Virus Reference Department, Public Health England, London, United Kingdom

Corresponding author
Carmen L. Sheppard,
Carmen.Sheppard@phe.gov.uk

## ABSTRACT

The major virulence factor of the pneumococcus, and target for conjugate vaccines, is the polysaccharide capsule, which is usually encoded by the highly variable *cps* locus. Serotype 37 is an unusual pneumococcal type in which the single β-glucosyltransferase gene responsible for serotype capsule production (*tts*) is located outside of the capsular operon region. Using a previously described automated whole genome sequence (WGS)-based serotyping bioinformatics tool, PneumoCaT, we identified and investigated seven clinical isolates (three from blood cultures) of non-pneumococcal streptococci containing a highly homologous *tts* and included them in a study panel of 20 isolates which included a 11 further clinical isolates of *S. pneumoniae* serotype 37, a reference strain of serotype 37 and the *S. pseudopneumoniae* type strain BAA 960[T]. The seven non-pneumococcal isolates generated novel alleles at all pneumococcal MLST loci and gave low percentage similarity (<45%) to *S. pneumoniae* or *S. pseudopneumoniae* species by comparison of short sequence patterns in genomic data (k-mer analysis). The *S. pseudopneumoniae* BAA-960[T] isolate generated two novel alleles in the MLST and gave a high similarity (>99%) to the reference sequence for BAA-960[T]. Twelve isolates gave high similarity (>77%) to the *Streptococcus pneumoniae* 5652-06 serotype 19A reference genome sequence and had previously reported MLST alleles. Each of the seven clinical non-pneumococcal strains and all of the 12 *S. pneumoniae* possessed a β-glycosyltransferase gene (*tts*) with >95% similarity to the pneumococcal *tts* reference DNA sequence with 20–22 non-synonymous SNPs. All but two strains in which the *tts* gene was detected gave positive reactions for serotype 37 in slide agglutination tests with serotype 37 typing sera. Phylogenetic analysis using both SNP and MLST data showed distinct clades corresponding to strains identified as pneumococcus or non-pneumococcus by kmer WGS analysis. Extended k-mer database analysis and ribosomal MLST placed the non-pneumococcal isolates within the *S. mitis* group. Biochemical and bile solubility assays showed differences between the unusual isolates and *S. pneumoniae*. All isolates had detectable pneumolysin (*ply*) genes, but only those that identified as pneumococcus contained the genes for autolysin (*lytA*) or the ABC
transporter lipoprotein A (*piaA*) with >80% coverage and >95% similarity. Here we report the existence of a novel group of strains distinct from *S. pneumoniae*, but which can express a pneumococcal serotype 37 capsular polysaccharide which can be associated with clinical disease.

## INTRODUCTION

The polysaccharide capsule of *Streptococcus pneumoniae* (pneumococcus) is an essential virulence factor (*Nelson et al., 2007*) and a distinguishing characteristic of the species compared to other closely related, non-encapsulated streptococci such as *S. pseudopneumoniae, S. oralis* and the *S. mitis* group streptococci. These other non-encapsulated species predominantly cause non-invasive disease, but can occasionally cause invasive diseases such as endocarditis and other infections in immunocompromised patients. They may also contain pneumococcus-like virulence genes such as autolysin (*lytA*) and pneumolysin (*ply*) leading to their potential misidentification as *S. pneumoniae* (*Whatmore et al., 2000*; *Balsalobre et al., 2006*; *Johnston et al., 2010*). The capsule of the pneumococcus plays a significant role in its pathogenesis and pneumococcal disease is a major global public health issue and cause of morbidity and mortality in young children and adults, in both developed and developing countries. The range of diseases caused by pneumococci includes severe manifestations, e.g., pneumonia, meningitis and bacteraemia, to less serious ones, such as otitis media, sinusitis and bronchitis (http://www.who.int/immunization/topics/pneumococcal_disease/en/). The capsule is also the target of all current licenced vaccines for *S. pneumoniae*, and the introduction of conjugate vaccines to the most common capsular types has led to a dramatic reduction in circulating vaccine serotypes and an increase in non-vaccine serotype disease (*Waight et al., 2015*).

In 90 of the 92 serotypes, defined in the Danish system by the commercial typing sera manufacturer Staten Serum Institute (SSI), Copenhagen, Denmark (http://www.ssi.dk/), the capsular polysaccharide is produced via the *wzx-wzy* dependent biosynthetic pathway, using a cluster of genes in an operon located between the *dexB* and *aliA* genes in the pneumococcal genome (*Bentley et al., 2006*). Serotypes 3 and 37 are unusual serotypes that utilise the synthase pathway for capsule production. In the synthase pathway, a single synthase gene produces an enzyme located in the cell membrane, which assembles and extrudes the simple polysaccharide chains (*Yother, 2011*). Serotype 37 pneumococcus has the gene for capsular polysaccharide production (*tts*) located outside of the capsular operon region (*Llull et al., 1999*), whereas in serotype 3, although a single gene (*wchE*) is also responsible, it is located within the regular capsular operon location (*Bentley et al., 2006*).

Serotype 37 pneumococci have a redundant, serotype 33F-like capsular operon in the region between *dexB* and *aliA*, which does not play a part in the production of serotype 37 capsular polysaccharide (*Llull et al., 1999*). Llull et al. described the transformation of other pneumococcal serotypes to a binary capsule type (that is expression of serotype 37 plus

another serotype) by insertion of the *tts* gene into other serotypes of pneumococcus. In further work, authors from the same laboratory demonstrated expression of the serotype 37 polysaccharide in other Gram-positive bacterial species, *S. oralis*, *S. gordonii* and *Bacillus subtilis* by laboratory transformation with plasmids containing the *tts* gene (*Llull, Garcia & Lopez, 2001*).

During development, validation and use of a bioinformatics pipeline for determining serotype in *S. pneumoniae* from whole genome sequencing (WGS) data, PneumoCaT (*Kapatai et al., 2016*), we discovered a number of isolates that were reported as *S. pneumoniae* serotype 37 using phenotypic methodology, but were subsequently identified as non-pneumococcal Streptococcus spp. by whole genome kmer analysis. They all contained a gene with high similarity to the serotype 37 capsule production gene, *tts*. We believe this to be the first description of non-pneumococcal isolates containing a *tts* β-glycosyltransferase gene and expressing serotype 37 polysaccharide, associated with clinical disease.

We found a total of seven non-pneumococcal streptococcal isolates identified by genomic analysis that each contained the *tts* gene. This study describes the characterisation of these isolates by WGS analysis and phenotypic methods.

## MATERIALS AND METHODS

### Study isolates

We created a study panel of 20 isolates, comprising 18 clinical isolates which were shown to contain the *tts* gene by PneumoCaT WGS analysis (described below), the serotype 37 *S. pneumoniae* type strain (SSI-37) and the *S. pseudopneumoniae* type strain (BAA-960$^T$). This panel of isolates was subjected to full characterisation using both phenotypic and genotypic methods.

Genomic data from an additional 17 *S. mitis* ($N = 5$), *S. pseudopneumoniae* ($N = 6$) and *S. pneumoniae* ($N = 6$, serotypes 23B, 14, 3, 22F, 8 and non-typeable) strains identified by kmerID, were also used for phylogenetic comparisons only, referred to as "extra isolates" in this manuscript.

Clinical isolates were received as part of the Public Health England (PHE) surveillance of pneumococcal infections in which referral of invasive *S. pneumoniae* (i.e., from blood and CSF) is currently requested from hospital laboratories in England and Wales to the Respiratory and Vaccine Preventable Bacteria Reference Unit (RVPBRU), London for serotyping. In addition, some pneumococci from non-sterile sites (e.g., sputum isolates) are also referred for antibiotic resistance testing and pass through RVPBRU for confirmation of ID and serotyping before being sent to the Antimicrobial Resistance and Healthcare Associated Infections Reference Unit for antibiotic resistance testing.

Isolates of *S. pseudopneumoniae* BAA-960$^T$ and the *S. pneumoniae* serotype 37 reference strain (SSI-37) were obtained from the American Type Culture Collection (ATCC), Manassas, VA, USA and Statens Serum Institut (SSI) Denmark, respectively.

All sequences from the study isolates that were identified as pneumococcus and one of the non-pneumococcal isolates are available on the European Nucleotide Archive as part

of our previous work (*Kapatai et al., 2016*), under project PRJEB14267, all other sequences used in this study are submitted to ENA under project number PRJEB20507, see Table S1.

## Phenotypic methods

Isolates were grown on Columbia blood agar (Oxoid, Chesire, UK) and had been putatively identified as *S. pneumoniae* by demonstration of optochin sensitivity using 5 µg optochin disc (Oxoid-Thermofisher, Basingstoke, UK). For phenotypic serotyping, isolates were grown for four hours or overnight in 5 ml MAST Todd Hewitt broth (PHE Media Services) at 35 °C with 5% $CO_2$, centrifuged at $453\times$ g for 30 min, the supernatant removed, the cell pellet re-suspended in a small residual volume of broth and serotyped by slide agglutination tests with standard antisera (SSI, Copenhagen, Denmark). The cell suspensions were also mixed with typing sera on a slide and methylene blue (Sigma, Welwyn Garden City, UK) was added. These were visualised under a light microscope using a $100\times$ oil immersion objective.

Additional biochemical testing was performed using API Rapid ID Strep 32 (bioMérieux, Basingstoke, UK) according to the manufacturer's instructions. Standard deoxycholate bile solubility test was performed by culture of the organism in duplicate test tubes containing 5 ml Todd-Hewitt broth. After centrifugation at $453\times$ g for 30 min the supernatants were discarded, cell pellets resuspended in 2.5 ml of phosphate-buffered saline (PBS), and 0.5 ml of 10% sodium deoxycholate added to one of the tubes (the other being the control). These were incubated at room temperature for 30 min. Complete lysis within 30 min of incubation in the tube containing the deoxycholate indicated a soluble result and incomplete lysis was scored as ''indeterminate'' (*Keith et al., 2006*).

## Electron microscopy

Electron microscopy using colony lifts was performed to visualise the general morphological characteristics of selected isolates. In this technique, 400 mesh Copper EM grids (Agar Scientific, Stansted, UK), coated in-house with pioloform (Agar Scientific, Stansted, UK) and carbon were placed directly onto the surface of bacterial colonies growing on blood agar and light pressure applied. After 1–2 min resting on the colonies, the EM grids were carefully removed and placed (cell-side-down) onto a drop of 3% (w/v) paraformaldehyde (PFA) in PBS. After 15 min incubation on the PFA, the grid was removed, its surface washed twice with deionized water and the bacterial cells negatively stained with 1.5% (w/v) phosphotungstic acid (Taab Laboratories). After drying the EM grids were examined in a JEM-1400 transmission electron microscope (JEOL, Welwyn Garden City, UK) and images acquired using an AMT XR60 digital camera (Deben, Tillbury, UK).

## DNA extraction and sequencing

Bacterial growth was harvested from the agar plates and pre-lysed by the recommended method for Gram-negative bacteria (Qiagen, Manchester, UK), which is effective for pneumococcus and closely related streptococci. DNA was extracted from the lysates using a QIAsymphony SP automated instrument and a QIAsymphony DSP DNA Mini Kit (Qiagen), using the tissue extraction protocol. DNA concentrations were measured using the Quant-IT Broad Range dsDNA Kit (Life Technologies, Paisley, UK) and GloMax® 96 Microplate Luminometer (Promega, Southampton, UK). WGS was performed using

Illumina methodology by the PHE Genomic Services Delivery Unit (GSDU, Colindale, UK). The resulting data were automatically analysed using an in-house bioinformatics pipeline for *S. pneumoniae.*

### PHE reference bioinformatics workflow for *S. pneumoniae*

Casava 1.8.2 (Illumina inc. San Diego, CA, USA) was used to deplex the samples and FASTQ reads were processed with Trimmomatic to remove adapters and bases from the trailing end that fall below a PHRED score of 30. K-mer identification software (kmerID, https://github.com/phe-bioinformatics/kmerid) was used to compare the sequence reads with a panel of curated NCBI RefSeq (https://www.ncbi.nlm.nih.gov/refseq/) genomes to identify the species. K-merID measures the identity between a reference genome and a set of WGS reads by determining the percentage of 18-mers in the reference genome that also occur at least twice in the WGS reads. Thus, each genomic sequence in the *S. pneumoniae* bioinformatics workflow was compared to 1769 reference genomes representing 59 pathogenic and commensal bacterial genera obtained from RefSeq. The closest percentage match was identified, and provided initial confirmation of the species. This step also identified isolates containing more than one species of bacteria (i.e., mixed cultures) and any organisms misidentified as *S. pneumoniae* by the sending laboratory. Further analysis (for MLST type and serotype) using the automated *S. pneumoniae* workflow continues only if *S. pneumoniae* was identified in the top five hits using the kmerID.

*Streptococcus pneumoniae* MLST data were derived from the genomic data using the Metric-Oriented Sequence Typing software (*Tewolde et al., 2016*; https://github.com/phe-bioinformatics/MOST) using allele definitions downloaded from PubMLST during the period Oct 2014 to December 2016. WGS serotype was derived from the data using a version of PneumoCaT (*Kapatai et al., 2016*; https://github.com/phe-bioinformatics/PneumoCaT) incorporated as the final analytic stage of the in-house *S. pneumoniae* WGS bioinformatics pipeline. Specific to this study, the PneumoCaT workflow initially mapped reads from the subject to a reference database containing capsular operon sequences for all serotypes, including serogroup 33 capsular operon sequences and the *tts* sequence. Due to the presence of the pseudo-33F capsular operon, pneumococcal isolates give >90% coverage for the 33F and 33A operon as well as for the *tts* gene. However, serotype 37 is determined by the presence of the *tts* gene (with >90% coverage and minimum depth of 5 reads per bp).

### Further genomic analysis methods

#### Analysis of the detected tts sequences

As PneumoCaT only provides coverage statistics for the *tts* genes detected, the NCBI BLAST website (https://blast.ncbi.nlm.nih.gov) was used to assist extraction of the *tts* sequence from the contigs and query the BLAST nucleotide collection database (nr/nt) and the BLAST protein collection database (pr) using algorithms blastn and blastp, respectively.

The amino acid explorer tools at the NCBI website (https://www.ncbi.nlm.nih.gov/Class/Structure/aa/aa_explorer.cgi) were used to assess the amino-acid substitutions observed between the study sequences and the reference TTS sequence and assess their likelihood of occurring in a homologous protein using the BLOSUM62 matrix (*Henikoff & Henikoff, 1992*).

### Evaluation of kmerID percentage similarity threshold for streptococcal species determination

KmerID uses the percentage of 18-mers in a given reference genome that are also present in a given set of WGS sequencing reads at least twice. In order to determine a percentage similarity threshold for the kmerID method for a positive identification of a species the Jaccard index of set similarity (JI) was used to investigate the relationship of 798 genomes belonging to 58 different streptococcal species to each other. Given two streptococcal genomes and the respective sets of 18-mers in these two genomes represented as A and B, the JI is calculated as the size of the intersection between A and B divided by the size of the union of A and B. For the 18 streptococcal species for which a minimum of five genomes are available, we compared the intra-species JIs with the inter-species indices and thus determined threshold values at which species can be reliably identified.

For this purpose, we iterated over a range of JI cut-off values and calculated the Matthew's correlation coefficients (MCCs) based on the resulting confusion matrices. Areas in which the MCC reaches 1.0 for a given species denote threshold values at and above which this species can be reliably distinguished from other species.

In order to assess the population structure of the reference sequences a Neighbour Joining tree was constructed for 798 streptococcal genomes using the JI as a measure of similarity.

### Extended streptococcal kmerID analysis

Sequence data from all the isolates in the study were re-analysed by kmerID as described above using an extended streptococcal reference database containing 798 streptococcal genomes of 58 different species (referred to as "extended streptococcal kmerID", list of species shown in Table S2).

### Ribosomal MLST

Genomes were assembled using SPAdes version 3.8.5 (*Bankevich et al., 2012*) and the resulting contigs used to query the 53 allele ribosomal MLST database (http://pubmlst.org/rmlst/) to determine *rps* alleles and obtain a ribosomal sequence type (rST) and information (including species identification) of isolates submitted to the database if an exact rST profile match was achieved. The database was also queried for speciation using the "Identify Species" link at the PubMLST rMLST website (http://pubmlst.org/rmlst/ accessed April 2016), in which the alleles are compared one by one, (*rpsA-rpmJ*) and the first *rps* allele sequence with an exact match in the database is reported along with the species of the isolate that contains the matching allele (*Jolley et al., 2012*).

### Whole genome SNP analysis

Genomic reads from the study set of isolates plus the extra sequences from *S. pneumoniae* isolates *of* different serotypes (serotype 23B, 8, 3, 14 and nontypeable) and *S. pseudopneumoniae* sequenced at PHE were mapped to the acapsular *S. pneumoniae* R6 reference sequence (NCBI accession number NC_003098) using BWA-MEM version 0.7.12 (*Li & Durbin, 2009*). Variants were called using GATK 2.6.5 (*McKenna et al., 2010*). Variants were then filtered to retain high quality SNPs based on the following conditions: depth of coverage (DP) $\geq$5, AD ratio (ratio between variant base and alternative bases) $\geq$0.8,

Mapping Quality (MQ) ≥30, ratio of reads with MQ0 to total number of reads ≤0.05. All positions that fulfilled the filtering criteria in >90% of the samples were joined to produce a multiple fasta format file where the sequence for each strain consists of the concatenated variants. This file was used as an input to generate a maximum likelihood (ML) tree using RAxML (*Stamatakis, 2014*) with the following parameters –m (substitutionModel) GTRCAT –b (bootstrapRandomNumberSeed) 12345 -# (numberOfRuns) 1000.

Distance matrices were constructed and group analysis was performed using MEGA 7 Software (*Kumar, Stecher & Tamura, 2016*).

### Multi-locus sequence analysis

Multi-Locus Sequence Analysis (MLSA) (*Hanage, Fraser & Spratt, 2006*; *Glaeser & Kämpfer, 2015*) was performed by extracting the MLST allele sequences for *aroE, gdh, gki, recP, spi* and *xpt* from the pileup files created during the MLST analysis using MOST and concatenated. The evolutionary history was inferred using the Minimum Evolution method (*Rzhetsky & Nei, 1992*) using MEGA 7 (*Kumar, Stecher & Tamura, 2016*) to construct a minimum evolution tree using 100 bootstraps. The trees also included concatenated sequences from the small panel of extra *S. pneumoniae, S. mitis* and *S. pseudopneumoniae* isolates obtained from the PHE collection for further comparison.

### Detection of lytA, ply and piaA genes

The widely used pneumococcal PCR target genes *lytA* and *ply* together with the additional gene *piaA*, were detected in the study set using a mapping-based approach using published bowtie 2 and Samtools software (*Langmead, 2010*), GenBank KP110770, HG531769 and AF338658.1:111,1,130 sequences respectively were used as references and a cut-off of 80% coverage and 95% identity were used to indicate a positive gene detection in this study.

## RESULTS

### Phenotypic results

The panel of 20 isolates was characterised using a variety of methods (Tables 1 and 2). A major and immediately obvious phenotypic difference between the pneumococcal and non-pneumococcal isolates in this study was that the pneumococcal isolates had a very mucoid appearance on the blood agar plates. In contrast, the seven clinical isolates that were non-pneumococcus (by kmerID) had small, non-mucoid colonies. However, they still demonstrated a glistening appearance and a smoother consistency compared to the dry, rough colonies of the *S. pseudopneumoniae* type strain (BAA-960$^T$).

Further biochemical analysis by bile solubility, API Strep 32 and serotyping gave the results shown in Table 1. The non-pneumococcal clinical isolates gave variable results in the API biochemical test, with most giving a biochemical profile result consistent with *S. oralis* ($N = 4$), *S. mitis* ($N = 2$) and *S. pneumoniae* ($N = 1$), they also gave resistant ($N = 1$) or indeterminate (incomplete clearing, $N = 6$) results with the bile solubility test. All the kmerID *S. pneumoniae* serotype 37 strains were also identified as *S. pneumoniae* by API 32 Strep and gave soluble results with the bile solubility test. The BAA-960$^T$ *S. pseudopneumoniae* type strain identified as *S. oralis* by API strep 32.

**Table 1** Phenotypic analysis of serotype 37 isolates, with BAA-960[T] as a *Streptococcus pseudopneumoniae* reference.

| Isolate | Colony appearance | Serotype by slide agglutination | Optochin sensitivity | Bile solubility | API 32 Strep profile | API species identification | API % |
|---|---|---|---|---|---|---|---|
| PHESPD0338 | Very Mucoid | 37 | Sensitive | Soluble | 72076741110 | *S. pneumoniae* | 88.0 |
| PHESPD0344 | Very Mucoid | 37 | Sensitive | Soluble | 40272741110 | *S. pneumoniae* | 99.9 |
| PHESPD0356 | Very Mucoid | 37 | Sensitive | Soluble | 46072741100 | *S. pneumoniae* | 77.5 |
| PHESPD0383 | Very Mucoid | 37 | Sensitive | Soluble | 60076741110 | *S. pneumoniae* | 99.9 |
| PHESPV0691 | Very Mucoid | 37 | Sensitive | Soluble | 76076741110 | *S. pneumoniae* | 87.4 |
| PHESPV1034 | Very Mucoid | 37 | Sensitive | Soluble | 40076741100 | *S. pneumoniae* | 99.6 |
| PHESPV1405 | Very Mucoid | 37 | Sensitive | Soluble | 60076741110 | *S. pneumoniae* | 99.9 |
| PHESPD0363 | Very Mucoid | 37 | Sensitive | Soluble | 46072741100 | *S. pneumoniae* | 77.5 |
| PHESPV0789 | Very Mucoid | 37 | Sensitive | Soluble | 40072601100 | *S. pneumoniae* | 97.9 |
| PHESPV1119 | Very Mucoid | 37 | Sensitive | Soluble | 50076741100 | *S. pneumoniae* | 99.9 |
| PHESPD0355 | Very Mucoid | 37 | Sensitive | Soluble | 50072541100 | *S. pneumoniae* | 99.9 |
| SSI-37 | Very Mucoid | 37 | Sensitive | Soluble | 52076741100 | *S. pneumoniae* | 98.3 |
| PHESPD0357 | Small | 37 | Sensitive | Insoluble | 00132541100 | *S. mitis* | 97.3 |
| PHENP00003 | Small | 37 | Sensitive | Indeterminate | 40036441100 | *S. oralis* | 95.3 |
| PHENP00005 | Small | 37 | Sensitive | Indeterminate | 40016441100 | *S. oralis* | 90.6 |
| PHENP00006 | Small | 37 | Sensitive | Indeterminate | 42052541100 | *S. pneumoniae* | 92.9 |
| PHENP00001 | Very Small | 37 | Sensitive | Indeterminate | 40116441120 | *S. oralis* | 94.6 |
| PHENP00002 | Very Small | Untypeable | Sensitive | Indeterminate | 40112441110 | *S. mitis* | 95.2 |
| PHENP00007 | Very Small | Untypeable | Sensitive | Indeterminate | 60016441100 | *S. oralis* | 96.0 |
| BAA-960[T] | Rough | Not done | Sensitive | Indeterminate | 64012441140 | *S. oralis* | 98.3 |

We were able to confirm serotype 37 by slide agglutination in five of the seven non-pneumococcal isolates (Table 1). Observation under the microscope (×100 oil immersion) also showed agglutination of the cells when type 37 serum (SSI) was applied to a bacterial suspension. Due to the very mucoid, non-cell wall associated, nature of the capsule in serotype 37 (and serotype 3), it is not usually possible to observe capsular swelling (Neufeld reaction) with this serotype and observation of agglutination is the usual indication of the reaction of the serotype 37 capsular polysaccharide with the typing sera (P. Landsbo Elverdal, SSI, pers. comm., 2016).

Electron microscopy of colony lifts (see Fig. 1) showed the unusual non-pneumococcal isolates had a distinct cell shape, being less rounded than the pneumococcal cells. There was a visible haze around the very mucoid pneumococcus and there appeared to be a clear area around the cells of the non-mucoid non-pneumococcus serotype 37. The non-encapsulated *S. pseudopneumoniae* type strain (BAA-960[T]) did not show this clear area.

## Genomic results

### Genomic analysis results using the standard PHE *S. pneumoniae* workflow

Genomic analysis of the panel of 20 study organisms using the standard PHE *S. pneumoniae* bioinformatics workflow (kmerID, MOST and PneumoCaT) are shown in Table 2.

One of the study isolates (PHESPD0357) gave a 'best percentage similarity'' by kmerID analysis with a *S. pneumoniae* reference, but with a low similarity of 41.0%. The other

**Table 2** Serotype 37 (*tts* gene detected) isolates with standard *Streptococcus pneumoniae* WGS workflow analysis results, including the *S. pseudopneumoniae* (BAA-960^T) and SSI serotype 37 strains for reference.

| Isolate number | Isolation site | MLST Sequence type | MLST profile (*aroE, gdh, gki, recP, spi, xpt, ddl*) | *tts* gene | kmerID top match reference genome | kmerID % similarity |
|---|---|---|---|---|---|---|
| PHESPD0338 | Blood | 447 | 29, 33, 19, 1, 36, 22, 31 | + | *Streptococcus pneumoniae* 5652-06 | 77.4 |
| PHESPD0344 | Blood | 447 | 29, 33, 19, 1, 36, 22, 31 | + | *Streptococcus pneumoniae* 5652-06 | 77.4 |
| PHESPD0356 | Blood | 447 | 29, 33, 19, 1, 36, 22, 31 | + | *Streptococcus pneumoniae* 5652-06 | 77.3 |
| PHESPD0383 | Blood | 447 | 29, 33, 19, 1, 36, 22, 31 | + | *Streptococcus pneumoniae* 5652-06 | 77.3 |
| PHESPV0691 | CSF | 447 | 29, 33, 19, 1, 36, 22, 31 | + | *Streptococcus pneumoniae* 5652-06 | 77.4 |
| PHESPV1034 | Blood | 447 | 29, 33, 19, 1, 36, 22, 31 | + | *Streptococcus pneumoniae* 5652-06 | 79.7 |
| PHESPV1405 | Blood | 447 | 29, 33, 19, 1, 36, 22, 31 | + | *Streptococcus pneumoniae* 5652-06 | 77.5 |
| PHESPD0363 | Blood | 447 | 29, 33, 19, 1, 36, 22, 31 | + | *Streptococcus pneumoniae* 5652-06 | 77.7 |
| PHESPV0789 | Unknown | 447 | 29, 33, 19, 1, 36, 22, 31 | + | *Streptococcus pneumoniae* 5652-06 | 77.9 |
| PHESPV1119 | Blood | 447 | 29, 33, 19, 1, 36, 22, 31 | + | *Streptococcus pneumoniae* 5652-06 | 77.5 |
| PHESPD0355 | Unknown | 447 | 29, 33, 19, 1, 36, 22, 31 | + | *Streptococcus pneumoniae* 5652-06 | 77.4 |
| SSI-37 | Unknown | 7243 | 192, 34, 19, 1, 36, 22, 445 | + | *Streptococcus pneumoniae* 5652-06 | 77.6 |
| PHESPD0357 | Blood | Novel | all loci novel | + | *Streptococcus pneumoniae* 5652-06 | 41.0 |
| PHENP00003 | Sputum | Novel | all loci novel | + | *Streptococcus pseudopneumoniae* IS7493 uid71153 | 34.6 |
| PHENP00005 | Sputum | Novel | all loci novel | + | *Streptococcus pseudopneumoniae* IS7493 uid71153 | 37.0 |
| PHENP00006 | Sputum | Novel | all loci novel | + | *Streptococcus pseudopneumoniae* IS7493 uid71153 | 36.4 |
| PHENP00001 | Sputum | Novel | all loci novel | + | *Streptococcus pseudopneumoniae* SK674 | 33.8 |
| PHENP00002 | Blood | Novel | all loci novel | + | *Streptococcus pseudopneumoniae* SK674 | 33.8 |
| PHENP00007 | Blood | Novel | all loci novel | + | *Streptococcus pseudopneumoniae* SK674 | 34.0 |
| BAA-960^T | Unknown | Novel | 139, 371, 345, *, 441, *, 656 | − | *Streptococcus pseudopneumoniae* IS7493 uid71153 | 99.8 |

**Notes.**
*novel allele.
CSF, cerebrospinal fluid.

unusual isolates gave a highest similarity by kmerID (all < 40%) with *S. pseudopneumoniae* (either reference strain SK674 or IS7493 uid74453; Table 2). Four of these isolates were from sputum (and had been referred to our laboratory for antibiotic resistance testing due to "unusual resistance" patterns) and three of these isolates had been obtained from blood cultures. (Table 2).

Within this panel, six of the seven unusual clinical isolates gave a similar pattern of unusual results in MLST (all 7 loci giving unrecognised alleles) and kmerID (identity <40% to a *S. pseudopneumoniae* reference). One isolate gave a kmerID match closer to a *S. pneumoniae* reference, but with identity of only 41%. Eleven other clinical isolates with phenotypic characteristics consistent with *S. pneumoniae* gave kmerID matches to the *S. pneumoniae* reference at 77.4–77.9% identity and were all the same MLST type, ST 447. The serotype 37 reference strain SSI-37 possessed a different (but related) MLST profile to the 11 *S. pneumoniae* isolates above and a 77.6% match with the *S. pneumoniae* reference by kmerID. The *S. pseudopneumoniae* type strain BAA-960^T gave two unrecognised and

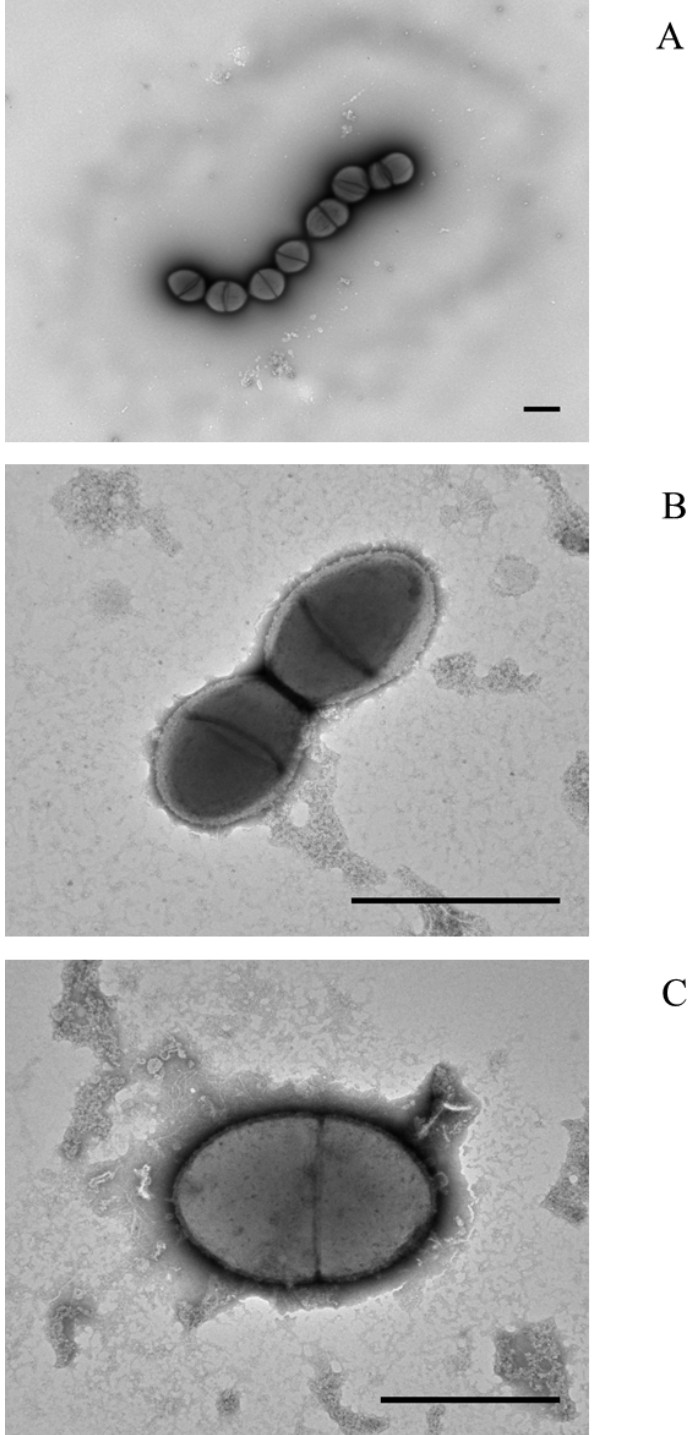

**Figure 1** Electron micrographs of (A) mucoid serotype 37 pneumococcus PHESPV1034, (B) unusual serotype 37 PHENP00003, (C) BAA-960 *Streptococcus pseudopneumoniae.* Scale bar 1 um.

five known alleles in the MLST analysis and a match of 99.8% identity to the BAA-960[T] reference sequence in the kmerID.

The presence of the *tts* gene was detected by mapping of raw WGS reads to a *tts* reference sequence within the PneumoCaT tool. All isolates, apart from BAA-960[T], had >98% coverage of the *tts* gene. BAA-960[T] had no coverage for *tts* (Table 2).

Although type 37 capsule production relies solely on the *tts* gene, *S. pneumoniae* serotype 37 isolates also possess an incomplete 33F-like capsular operon in their genome (*García, Llull & López, 1999*). The mapping coverage data obtained from the PneumoCaT tool (in which WGS reads from each isolate were mapped to all 92 reference pneumococcal capsular operon sequences as the first stage of the serotype distinction analysis) were used to determine whether the unusual isolates contained regions of known capsular operons. The results are shown in Table S3. The results revealed that the seven non-pneumococcal isolates contained DNA sequences that showed coverage for small regions of different capsular operons, but none demonstrated >20% coverage for any of the reference sequences, and the *S. pseudopneumoniae* reference strain BAA-960[T] showed 26.4% coverage to serotype 36 capsular operon (Table S3) and so a pneumococcus-like capsular operon was judged to be absent in these isolates. In comparison, a representative pneumococcal serotype 37 isolate, PHESPV1034, demonstrated >96% coverage of the 33F or 33A capsule operon.

### Analysis of the tts sequences

Further analysis of the 1530bp *tts* gene sequences in 19 of the 20 study isolates (i.e., excluding BAA-960[T]) revealed that, although there was some nucleotide sequence variation in the study set, NCBI BLAST queries using megablast (optimized for sequences with high-similarity) only returned one hit for all organisms, which was the same *S. pneumoniae* sequence used as the PneumoCaT *tts* reference (GenBank AJ131985.1 positions 2146–3675). BLASTn (optimized for somewhat similar sequences) returned the same sequence as top hit as expected. Translated protein sequences of 509 amino acids could be divided into four main groups. All the pneumococcal isolates except the SSI-37 matched the translated reference sequence (GenBank AJ131985.1 positions 2146–3675) exactly (see Table 3). The SSI-37 had a unique sequence, with a single amino acid tryptophan -> glycine difference at amino acid position 341 in the protein sequence compared to the 18 others studied (and the reference sequence), which all contained tryptophan at this position. The non-pneumococcal isolates all had the same sequences and only 20 amino acid differences throughout the gene compared to the reference sequence. However, two isolates also contained a DNA insertion in an AT repeat region that led to addition of an extra isoleucine and tyrosine amino acid residue at positions 85 and 86 in the protein sequence for these two isolates.

Analysis of the amino acid substitutions showed that there were five positions in the protein sequence where the non-pneumococcal isolates had amino acid substitutions with negative BLOSUM62 scores (Table 3). This score indicates that this amino acid substitution would be expected to occur only rarely in a homologous protein and may indicate a change in the structure or function of the protein (i.e., the proteins are non-homologous). These substitutions were as follows: position 99 A →D, 247 Y →R, 389 V →F, 473 T →I and

**Table 3** **Sites within the TTS protein sequence that show amino acid variability for 19 study panel isolates (with amino acid codes), including BLOSUM62 score for likelihood of substitution in a homologous protein.**

| Isolate | Amino acid position | | | | | | | | | | | | | | | | | | | | | | |
|---|---|---|---|---|---|---|---|---|---|---|---|---|---|---|---|---|---|---|---|---|---|---|---|
| | 17 | 79 | 82 | 85[a] | 86[a] | 88 | 89 | 99 | 118 | 126 | 243 | 247 | 341 | 348 | 376 | 389 | 416 | 463 | 473 | 476 | 479 | 492 | 509 |
| *tts* reference | S | N | H | – | – | S | S | A | V | A | K | Y | W | V | F | V | V | L | T | T | I | A | V |
| PHESPD0338 | S | N | H | – | – | S | S | A | V | A | K | Y | W | V | F | V | V | L | T | T | I | A | V |
| PHESPD0344 | S | N | H | – | – | S | S | A | V | A | K | Y | W | V | F | V | V | L | T | T | I | A | V |
| PHESPD0356 | S | N | H | – | – | S | S | A | V | A | K | Y | W | V | F | V | V | L | T | T | I | A | V |
| PHESPD0383 | S | N | H | – | – | S | S | A | V | A | K | Y | W | V | F | V | V | L | T | T | I | A | V |
| PHESPV0691 | S | N | H | – | – | S | S | A | V | A | K | Y | W | V | F | V | V | L | T | T | I | A | V |
| PHESPV1034 | S | N | H | – | – | S | S | A | V | A | K | Y | W | V | F | V | V | L | T | T | I | A | V |
| PHESPV1405 | S | N | H | – | – | S | S | A | V | A | K | Y | W | V | F | V | V | L | T | T | I | A | V |
| PHESPD0363 | S | N | H | – | – | S | S | A | V | A | K | Y | W | V | F | V | V | L | T | T | I | A | V |
| PHESPV0789 | S | N | H | – | – | S | S | A | V | A | K | Y | W | V | F | V | V | L | T | T | I | A | V |
| PHESPV1119 | S | N | H | – | – | S | S | A | V | A | K | Y | W | V | F | V | V | L | T | T | I | A | V |
| PHESPD0355 | S | N | H | – | – | S | S | A | V | A | K | Y | W | V | F | V | V | L | T | T | I | A | V |
| SSI-37 | S | N | H | – | – | S | S | A | V | A | K | Y | G | V | F | V | V | L | T | T | I | A | V |
| PHESPD0357 | T | K | Y | – | – | N | N | D | A | V | N | R | W | I | Y | F | L | F | I | M | F | V | L |
| PHENP00003 | T | K | Y | – | – | N | N | D | A | V | N | R | W | I | Y | F | L | F | I | M | F | V | L |
| PHENP00005 | T | K | Y | – | – | N | N | D | A | V | N | R | W | I | Y | F | L | F | I | M | F | V | L |
| PHENP00006 | T | K | Y | – | – | N | N | D | A | V | N | R | W | I | Y | F | L | F | I | M | F | V | L |
| PHENP00007 | T | K | Y | – | – | N | N | D | A | V | N | R | W | I | Y | F | L | F | I | M | F | V | L |
| PHENP00001 | T | K | Y | I | Y | N | N | D | A | V | N | R | W | I | Y | F | L | F | I | M | F | V | L |
| PHENP00002 | T | K | Y | I | Y | N | N | D | A | V | N | R | W | I | Y | F | L | F | I | M | F | V | L |
| BLOSUM62 Score[b] | 1 | 0 | 2 | n/a | n/a | 1 | 1 | −2 | 0 | 0 | 0 | −2 | −2 | 2 | 2 | −1 | 1 | 0 | −1 | −1 | 0 | 0 | 1 |

**Notes.**

[a]Amino acids insertions. All other stated amino acid positions are referenced according to the sequence in isolates that do not contain the insertions; n/a. not applicable.

[b]Residues in with positive scores (green) substitute frequently in homologous proteins (positive BLOSUM62 score), while those in pink substitute rarely.

476 T →M. The amino acid difference seen in the SSI-37 reference strain (341 W →G) also had a negative BLOSUM62 score. The properties of the amino acids which could potentially cause functional differences are shown in Table S4.

BLASTp search of the protein sequence for each of the groups showed that all had 9/10 top hits with *S. pneumoniae* glycosyltransferase genes.

### Evaluation of kmerID %similarity threshold for streptococcal species determination

The results of the threshold analysis intended to inform the selection of a cut-off for the kmerID, are shown in Table S5. The results showed that the chosen 65% (representing 0.65 JI value) is a conservative cut-off for a match to *S. pneumoniae* in the kmerID method, which possibly allows for some false-negative classification as non-pneumococcus if an isolate were to demonstrate kmerID similarity of between 60 and 65%.

The neighbour joining tree of all the reference strains used in the kmerID threshold analysis showed *S. pseudopneumoniae* as an out-group to *S. pneumoniae*. *Streptococcus mitis* was most closely related to the pneumococcal/*S. pseudopneumoniae* branch, but with overall low intra-species similarity (Fig. S1).

### Extended kmerID analysis and rMLST

We analysed the genomic data from the isolates in the study panel against an extended streptococcal kmerID database (798 genomes covering 58 streptococcal species) and uploaded assemblies to query the BIGSdb rMLST website for rMLST type and speciation matches. The results of these analyses are shown in Table 4.

Results of the extended kmerID database analysis on the 12 pneumococcal serotype 37 strains (including SSI-37) showed the highest percentage identity (range 80.8–83.2%) to the *S. pneumoniae* GA19690 reference genome. Seven *S. pneumoniae* isolates gave the same ribosomal sequence type (rST) 23,921. Four *S. pneumoniae* clinical isolates, the SSI-37 type strain and all of the non-pneumococcal species gave a novel rST. The number of recognized alleles varied between isolates.

Six of the seven non-pneumococcal isolates were identified only as *Streptococcus* spp. using the rMLST database speciation tool. Analysis with extended kmerID showed these six gave closest identity to the *S. mitis* SK1080 genome reference, but as seen with the *S. mitis* reference sequences the identity was low (range 37–38%).

The remaining non-pneumococcal isolate showed a different identification pattern to the others, identifying more closely with *S. pneumoniae* by both rMLST speciation (*rpsC*) and both the original and extended kmerID database, although against the extended kmerID database it identified closest (41.3% identity) to a different *S. pneumoniae* reference (GA47688) than the pneumococcal isolates in the study panel.

### Whole genome SNP analysis

A maximum likelihood phylogenetic tree of the study set isolates plus extra comparison strains was constructed. The tree was inferred from the variant alignment derived from SNP variant analysis using the R6 strain as reference. The analysis involved 36 nucleotide sequences. All positions with less than 90% site coverage were eliminated. That is, fewer than

**Table 4  Further genomic analysis, extended kmerID database and rMLST.**

| Isolate | PubMLST BIGSdb rMLST | | | | | Extended kmerID | |
|---|---|---|---|---|---|---|---|
| | rST | Profile match species | No. Alleles matched | Speciation match (single *rps* allele) | Speciation first match gene | Top match reference genome | Similarity % |
| PHESPD0338 | 23,921 | *S. pneumoniae* | 52 | *S. pneumoniae* | *rpsA* | *S. pneumoniae* GA19690 | 81.1 |
| PHESPD0344 | 23,921 | *S. pneumoniae* | 52 | *S. pneumoniae* | *rpsA* | *S. pneumoniae* GA19690 | 81.1 |
| PHESPD0356 | 23,921 | *S. pneumoniae* | 52 | *S. pneumoniae* | *rpsA* | *S. pneumoniae* GA19690 | 80.8 |
| PHESPD0383 | 23,921 | *S. pneumoniae* | 52 | *S. pneumoniae* | *rpsA* | *S. pneumoniae* GA19690 | 80.9 |
| PHESPV0691 | 23,921 | *S. pneumoniae* | 52 | *S. pneumoniae* | *rpsA* | *S. pneumoniae* GA19690 | 81.1 |
| PHESPV1034 | 23,921 | *S. pneumoniae* | 52 | *S. pneumoniae* | *rpsA* | *S. pneumoniae* GA19690 | 83.2 |
| PHESPV1405 | 23,921 | *S. pneumoniae* | 52 | *S. pneumoniae* | *rpsA* | *S. pneumoniae* GA19690 | 81.1 |
| PHESPD0363 | novel | – | 51 | *S. pneumoniae* | *rpsA* | *S. pneumoniae* GA19690 | 81.2 |
| PHESPV0789 | novel | – | 51 | *S. pneumoniae* | *rpsA* | *S. pneumoniae* GA19690 | 81.1 |
| PHESPV1119 | novel | – | 50 | *S. pneumoniae* | *rpsA* | *S. pneumoniae* GA19690 | 81.1 |
| PHESPD0355 | novel | – | 51 | *S. pneumoniae* | *rpsA* | *S. pneumoniae* GA19690 | 81.1 |
| SSI-37 | novel | – | 48 | *S. pneumoniae* | *rpsA* | *S. pneumoniae* GA19690 | 81.3 |
| PHESPD0357 | novel | – | 37 | *S. pneumoniae* | *rpsC* | *S. pneumoniae* GA19690 | 41.3 |
| PHENP00003 | novel | – | 47 | *Streptococcus* spp. | *rpsB* | *S. mitis* SK1080 | 36.9 |
| PHENP00005 | novel | – | 39 | *Streptococcus* spp. | *rpsB* | *S. mitis* SK1080 | 37.1 |
| PHENP00006 | novel | – | 40 | *Streptococcus* spp. | *rpsB* | *S. mitis* SK1080 | 38.0 |
| PHENP00001 | novel | – | 48 | *Streptococcus* spp. | *rpsA* | *S. mitis* SK1080 | 37.1 |
| PHENP00002 | novel | – | 48 | *Streptococcus* spp. | *rpsA* | *S. mitis* SK1080 | 37.4 |
| PHENP00007 | novel | – | 52 | *Streptococcus* spp. | *rpsA* | *S. mitis* SK1080 | 37.5 |
| BAA-960[T] | 31,046 | *S. pseudopneumoniae* | 52 | *S. pseudopneumoniae* | *rpsA* | ND | ND |

**Notes.**
–, no match;  ND,  Not Done.

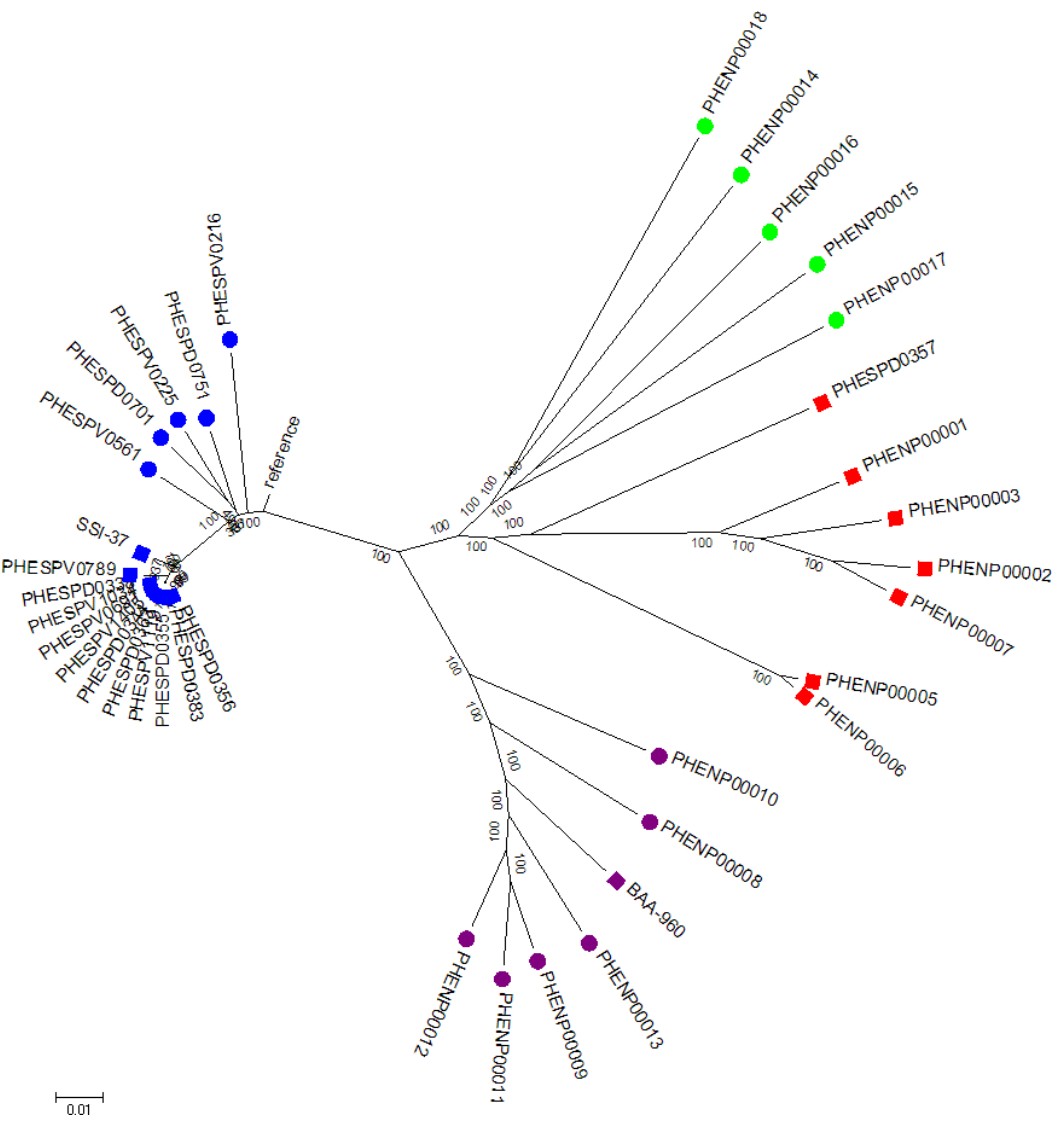

**Figure 2** **Whole genome SNP analysis maximum likelihood tree (RAxML bipartition) of study set isolates using the acapsular R6 reference.** Blue squares represent study set of *S. pneumoniae* isolates, purple squares the *S. pseudopneumoniae* type strain and red squares the study set non-pneumococcal isolates. Additional contextual data were provided by *Streptococcus pneumoniae* belonging to different serotypes (*N* = 5, blue circles), *S. pseudopneumoniae* (*N* = 6, purple circles) and *S. mitis* strains (*N* = 5, green circles). Branch length corresponds to substitutions per base. Bootstrap support shown for all branches.

10% alignment gaps, missing data, and ambiguous bases were allowed at any position. There were a total of 63,939 positions in the final dataset. The results showed that whole genome SNP analysis also separated the two groups of *S. pneumoniae* and non-pneumococcal isolates that contain the *tts* gene (Fig. 2). The non-pneumococcal isolates had a common ancestor to the *S. mitis* strains, but formed a separate branch with strong bootstrap support. Nucleotide differences within and between the groups of organisms forming the four main branches on the tree (*S. pneumoniae*, *S. pseudopneumoniae*, *S. mitis* and non-pneumococcal *tts*-positive) showed that the inter-group differences were greater than 5,000 SNPs

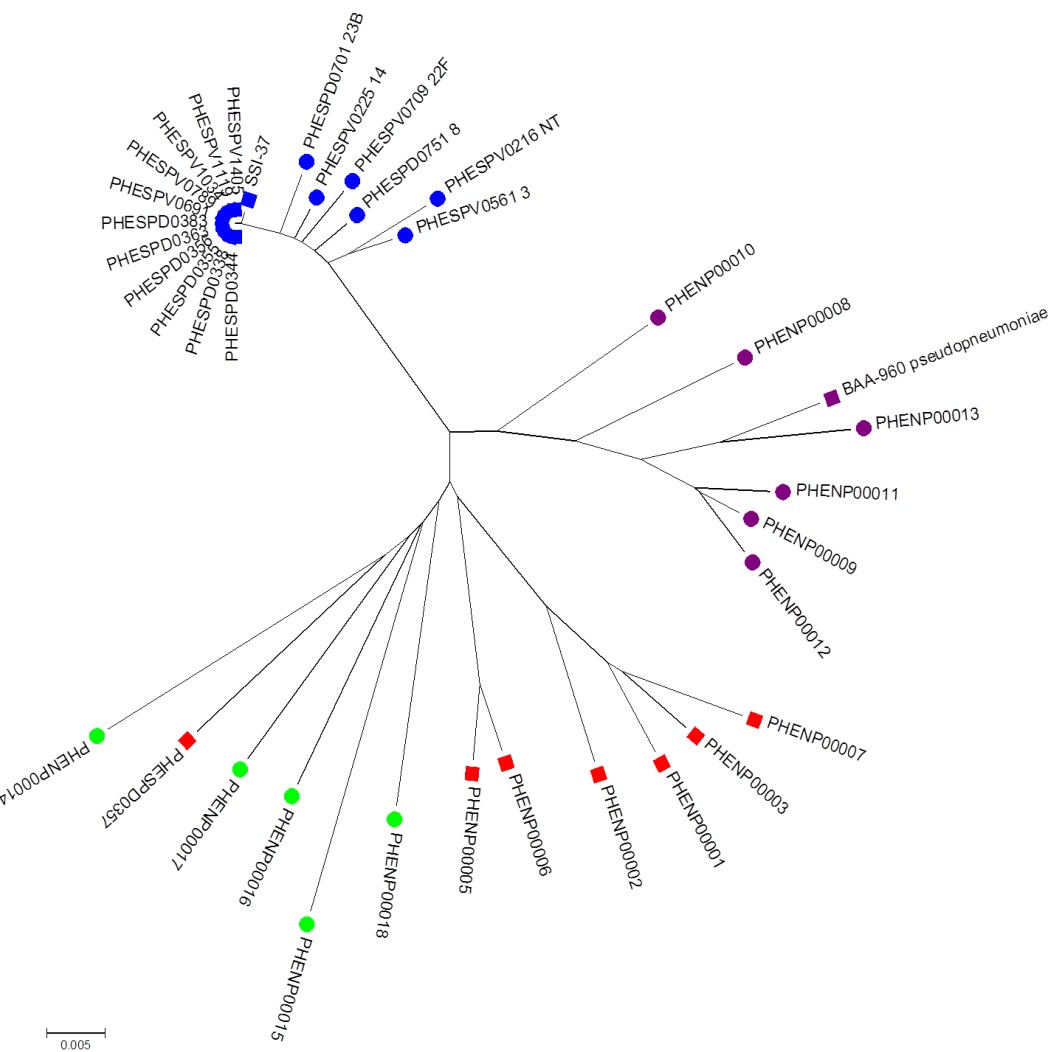

**Figure 3** **Minimum evolution tree of concatenated MLST sequences (except *ddl*) of study set isolates with contextual isolates.** Blue squares represent *S. pneumoniae* isolates, purple squares the *S. pseudopneumoniae* type strain and red squares non-pneumococcal isolates included in this study. Additional contextual data from other *Streptococcus pneumoniae* serotypes (blue circles), *S. pseudopneumoniae* (purple circles) and *S. mitis* strains (green circles).

(range 5,858–7,662). The groups showed variable within-group differences, the *S. pneumoniae* and *S. pseudopneumoniae* groups showed fewest SNP differences within group (1,545 and 3,556 respectively) and the *S. mitis* and the non-pneumococcal *tts* positive isolates showed the greatest number of within group SNP differences (7,811 and 5,327 respectively). The distance table is shown in Table S6.

### Multi-locus sequence analysis

The minimum evolution tree drawn using concatenated MLST allele sequence data for concatenated *aroE*, *gdh*, *gki*, *recP*, *spi*, and *xpt* sequences is shown in Fig. 3. As they were all the same MLST type, all the clinical pneumococcal serotype 37 isolates in the study set formed a single branch on the tree with the SSI-37 type strain forming a separate, but closely

**Table 5** *lytA*, *ply* and *piaA* gene detection, coverage and identity statistics for isolates in the study.

| Isolate | *lytA* coverage % | *lytA* % identity to KP110770 | *ply* coverage % | *ply* identity to HG531769 | *piaA* coverage % | *piaA* identity to AF338658.1:111–1,130 |
|---|---|---|---|---|---|---|
| PHESPD0338 | 100 | 99.58 | 100 | 99.65 | 100 | 99.6 |
| PHESPD0344 | 100 | 99.58 | 100 | 99.65 | 100 | 99.6 |
| PHESPD0356 | 100 | 99.58 | 100 | 99.65 | 100 | 99.6 |
| PHESPD0383 | 100 | 99.58 | 100 | 99.65 | 100 | 99.6 |
| PHESPV0691 | 100 | 99.58 | 100 | 99.65 | 100 | 99.6 |
| PHESPV1034 | 100 | 99.58 | 100 | 99.65 | 100 | 99.6 |
| PHESPV1405 | 100 | 99.58 | 100 | 99.65 | 100 | 99.6 |
| PHESPD0363 | 100 | 99.58 | 100 | 99.65 | 100 | 99.6 |
| PHESPV0789 | 100 | 99.58 | 100 | 99.65 | 100 | 99.6 |
| PHESPV1119 | 100 | 99.58 | 100 | 99.65 | 100 | 99.6 |
| PHESPD0355 | 100 | 99.58 | 100 | 99.65 | 100 | 99.6 |
| SSI-37 | 100 | 99.58 | 100 | 99.65 | 100 | 99.6 |
| PHESPD0357 | 88 | 73.25 | 100 | 96.96 | 0 | 0 |
| PHENP00003 | 85.8 | 76.28 | 100 | 97.1 | 0 | 0 |
| PHENP00005 | 92.7 | 78.16 | 100 | 95.41 | 0 | 0 |
| PHENP00006 | 93.3 | 77.32 | 100 | 95.34 | 0 | 0 |
| PHENP00001 | 78.2 | 67.92 | 100 | 97.03 | 0 | 0 |
| PHENP00002 | 86.3 | 75.03 | 100 | 96.75 | 0 | 0 |
| PHENP00007 | 87.4 | 75.44 | 100 | 96.19 | 0 | 0 |
| BAA-960[T] | 91.8 | 78.47 | 100 | 96.68 | 0 | 0 |

related branch. The additional *S. pneumoniae* sequences of various other serotypes added to the analysis formed closely related, but separate branches within the same sub-branch of the tree as the study pneumococcal isolates. The additional *S. pseudopneumoniae* and *S. mitis* sequences added to the analysis formed separate clusters on the tree, with the study set of non-pneumococcal sequences forming their own sub-branch near the *S. mitis* cluster except for the PHESPD0357 isolate, which also produced differing results on initial kmerID and on the rMLST and clustered closer within the *S. mitis* isolates on the tree rather than on the separate branch.

### *lytA, ply* and *piaA* gene detection

Detection and analysis of the *lytA*, *ply* and *piaA* genes revealed that all the isolates in the study contained genes with identity to the reference sequences used for *lytA* and *ply*. However, none of the non-pneumococcal isolates contained any sequences homologous to the *piaA* reference sequence or passed the cut-off for detection of *lytA* (80% coverage and 95% identity). Hence, the *piaA* and *lytA* genes were classed as 'not detected' in the non-pneumococcal isolates, whereas the *ply* gene was detected in all isolates. The coverage and identity statistics are shown in Table 5.

## DISCUSSION

The use of WGS technology in conjunction with bioinformatics analysis will continue to revolutionise reference microbiology and our understanding of bacterial evolution due to the enhanced discrimination of the methodology, and will likely re-write the phylogeny of the bacterial domain. However, how to classify recombinant bacteria with their fluid genomes has been a subject of considerable debate; historically, bacteriologists named a species according to common phenotypic traits such as morphology and sugar fermentation (*Winslow et al., 1917*). In this genomic era, however, it is now commonly discussed if genomic similarity should be the species determinant and whether determining a species is, in fact, possible for fluid bacterial genomes. The Core Genome Hypothesis has been proposed to explain why fluid bacterial genomes can have stable clusters (*Riley & Lizotte-Waniewski, 2009*) which could be named as species. In this hypothesis it is proposed that core genes are responsible for maintaining the phenotypically distinct species clusters.

In this study, we evaluated both phenotypic and genomic data in an attempt to characterise a number of unusual streptococcal clinical isolates that were discovered in our laboratory using our standard WGS workflow for *S. pneumoniae*. The workflow includes the kmerID method, MLST and capsular typing, and the combination of these three techniques allowed the recognition of clinically relevant and potentially novel streptococcal isolates that have the *tts* gene for production of serotype 37 capsular polysaccharide, but which were not *S. pneumoniae*.

SNP analysis and MLST sequence analysis identified these organisms as being similar to, but distinct from isolates identified as *S. pneumoniae* or *S. pseudopneumoniae*, but most closely related to *S. mitis* isolates. As described previously (*Hanage, Fraser & Spratt, 2006*) and in our own study, *S. mitis* shows greater genetic diversity than those organisms defined as *S. pneumoniae* or *S. pseudopneumoniae*. Analysis using our kmerID tool to measure whole genome similarity between reference genomes (Fig. S1) showed that *S. mitis* reference genomes do not co-locate in a distinct branch like other streptococcal species. Instead, individual genomes are separated by long branch lengths. The branch lengths were as long between *S. mitis* reference genomes as those defining the other species (e.g.. *S. pneumoniae*, *S. pseudopneumoniae*) in the tree. This was also seen in the analysis of clinical isolates using both whole genome SNP analysis (Fig. 2) and MLSA (Fig. 3). These data do not support the designation of *S. mitis* as a species, originally defined by serological and biochemical methods (*Facklam, 1977*). For that reason we can only define the unusual isolates seen in this study as "most similar to *S. mitis*" given that isolates biochemically designated as *S. mitis* are also only " most similar to *S. mitis*" when studied by genomic methods rather than confidently placed within a species branch like *S. pneumoniae* or *S. pseudopneumoniae*.

Presence of the *tts* gene has not been previously reported in clinical non-pneumococcal isolates to our knowledge. *Llull et al. (1999)* showed that introduction of the *tts* gene to other pneumococcal serotypes and even other Gram-positive organisms *in vitro* caused a binary capsule in pneumococcal isolates containing a capsular operon (*García, Llull & López, 1999*) and that the *tts* gene alone is responsible for capsular production in

organisms without a capsular operon (*Llull, Garcia & Lopez, 2001*). This was supported by the detection of serotype 37 polysaccharide in five isolates in this study which lacked a capsular operon.

Despite the phylogenetic variation in the group of non-pneumococcal *tts* positive organisms in this study, the TTS protein sequence was identical, therefore it is likely that this gene was transferred by horizontal gene transfer, and evolved and differentiated from that seen in pneumococcus at a point in the past.

Although there were some amino acid substitutions that are only rarely seen in homologous protein sequences according to the BLOSUM62 Matrix, many of these appear to only give slight differences in the amino acid properties. Based on the discussion in *Llull, Garcia & Lopez (2001)* the TTS protein contains six transmembrane helices, five of the amino acid substitutions we found in the non-pneumococcal isolates are within these transmembrane helix regions. Only one of these five amino acids had a negative BLOSUM62 score; at position 389, valine was substituted with phenylalanine. These two amino acids differ only by an aromatic ring but share the same hydrophobicity score (0.923 and 0.951 respectively), polarity and lack of H-bond formation, and they both prefer to adopt β-strand conformations.

The difference at residue 247, where tyrosine is present in pneumococcal TTS sequence and arginine in the non-pneumococcal sequence, is the change with the most impact on amino acid properties, from a relatively hydrophobic amino acid in the pneumococcal protein to a very hydrophilic one.

It is unlikely that these differences in sequence would dramatically change the function of the TTS protein compared to that found in *S.pneumoniae*, but this could only be confirmed by full structural and functional analysis.

The TTS amino acid sequence appears to be stable and it therefore not a recent acquisition by one particular clone of these organisms. There is no evidence of clonal expansion of these *tts* positive organisms in the isolates we have found.

Further investigation is needed to determine the regulation of the *tts* gene expression and polysaccharide production in the organisms characterised in this study, as there was clearly a difference between the capsular expression for the *S. pneumoniae* serotype 37 isolates in which the capsule is abundant and mucoid, and the unusual isolates which appear to express far less polysaccharide and have a non-mucoid appearance. It is possible some parts of the redundant regular capsular operon or other external factors may play some part in the augmentation of expression of the phenotype in the pneumococcus, and certainly these factors warrant further study. Non-encapsulated pneumococci and non-pneumococcal streptococci play a major role in horizontal transfer of genetic information, their acapsular phenotype allowing more ready interchange of genetic information than capsulated organisms (*Chewapreecha et al., 2014*). The seemingly limited expression of the polysaccharide in these *tts* positive non-pneumococcal organisms could mean transfer of genetic information may be more likely than in the pneumococcus, leading to this version of the *tts* gene being found in more diverse organisms.

These isolates appear to be rare, mostly non-invasive and may not normally be identified as *S. pneumoniae* on the bench of a routine microbiology laboratory. Therefore, typically

they would not be referred for serotyping to a reference laboratory such as the PHE *S. pneumoniae* reference laboratory for serotyping and thus they would remain unidentified. Even if referred to reference laboratories such isolates may also be misidentified as pneumococcal serotype 37 or non-typeable organisms unless WGS methods are used.

In the future, thorough WGS analyses for pneumococcal serotype, which include analysis of the entire capsular operon and necessary related genes (such as *tts*), should be more sensitive to the detection of potentially novel/mixed results (e.g., binary capsules, should they exist *in vivo*), than methods that focus on smaller regions of the sequence. A recent study of capsular production in non-pneumococcal species such as *S. mitis* and *S. oralis* showed that the production of capsule via the *wzx-wzy* pathway in these species is not uncommon and some bear antigenic and genetic similarity to those recognised for *S. pneumoniae* which could lead to misidentification of these species (*Skov Sørensen et al., 2016*). Previous studies have also shown an abundance of pneumococcal capsular gene homologs present in non-pneumococcal streptococci which can confound PCR based serotyping methods (*Carvalho et al., 2013*) However, we have not yet found any examples of misidentified isolates of this nature in the clinical isolates we have studied to date using our WGS methods which should, as seen with the serotype 37 isolates, be sensitive enough to recognise any previously misidentified isolates by combined detection of capsular operon and use of whole genome kmer analysis for species identification.

For laboratory identification of these isolates, which could be potentially misidentified as pneumococcal serotype 37, DNA sequence-based methods were superior to phenotypic methods for obtaining a clear differentiation between *S. pneumoniae* and closely related species. Athough *lytA* was defined as 'not-detected' in the non-pneumococcal isolates using our coverage and identity cut-off, the presence of both *lytA*-like and *ply* sequences in all of these strains may confound species identification methods based on short-sequence detection such as PCR or oligonucleotide probe hybridization, however the alternative target, *piaA*, which has previously been suggested as a specific pneumococcal PCR target (*Whalan et al., 2006*; *Trzciński et al., 2013*) does appear to offer species differentiation in these organisms.

Currently the standard *S. pneumoniae* seven allele MLST scheme offers a suitable alternative to WGS for determining whether an isolate is a pneumococcus or not. In our study, the presence of multiple unrecognised alleles (>1) in the standard pneumococcal MLST scheme (PubMLST database accessed October 2016) correlated with the WGS kmerID identification as non-pneumococcal species. When the whole genome analysis gave a result of <65% identity to *S. pneumoniae* by kmerID, the MLST also gave multiple unrecognised alleles (>1 locus). This was also supported by the use of the concatenated sequence data from six of the alleles in the MLSA analysis (excluding *ddl*) which was useful for producing phylogenetic trees and may also be used to help assign an identity if comparative species sequences are available. If the presence of multiple novel alleles in the standard MLST data is to be used as an indication of a non-pneumococcal species, it is important that the database of MLST alleles for *S. pneumoniae,* hosted at PubMLST, is kept free of sequences from isolates misidentified as pneumococcus. Up to the time of our study, well-managed curation of the *S. pneumoniae* MLST database appears to

have excluded allele sequences that show excessive variation from standard *S. pneumoniae* sequences, thus suggesting non-pneumococcal origin, and has so far enabled the database to be useful in determining whether an isolate is a pneumococcus or not simply by the number of novel alleles seen in the organisms.

Limited clinical information was available for the non-pneumococcal *tts* positive isolates. Of the three isolates that were obtained from blood cultures, two of the patients had pneumonia and one bacteraemia. Two of the isolates from blood cultures came from patients with co-morbidities (immunosuppression in one and congenital mitochondrial cytopathy in another), suggesting that although they can be seen in invasive disease these organisms may also be opportunistic and require more susceptible hosts to enable invasion to normally sterile sites. Clinical details are shown in Table S7.

The abundance of these non-pneumococcal species containing the *tts* gene and their potential clinical relevance compared to pneumococcus is unknown and they are likely to be often misidentified in the routine laboratory. However, the increasing use of molecular and, in particular, whole genome analysis techniques in reference laboratories should increase the likelihood of identification of these organisms. This will allow further opportunities to study their molecular and clinical characteristics and enable the further description or classification of fluid bacterial species like those in the *S. mitis* group.

## ACKNOWLEDGEMENTS

The authors would like to thank our colleagues at PHE, London UK, including McDonald Prest for retrieving isolates from storage, DNA extraction and performing bile solubility tests, Ella Campion, Gurkiran Mankoo, Sophie Hang, and John Duncan for serotyping isolates, Richard Myers, for discussion on kmerID and phylogenetics and Sarah Collins of the Immunisation Department for providing matched clinical information. We would like to thank Pernille Landsbo Elverdal (SSI Diagnostica, Denmark), for discussions on serotyping of 37 isolates. We would also like to thank Bill Hanage (Harvard University, USA) who participated in helpful discussions on MLSA.

### Funding

The authors received no funding for this work.

### Competing Interests

The authors declare there are no competing interests.

### Author Contributions

- Carmen L. Sheppard, Georgia Kapatai, Ulf Schaefer and Matthew Hannah conceived and designed the experiments, performed the experiments, analyzed the data, contributed reagents/materials/analysis tools, wrote the paper, prepared figures and/or tables, reviewed drafts of the paper.

- Karen Broughton performed the experiments, contributed reagents/materials/analysis tools, reviewed drafts of the paper.
- David J. Litt and Norman K. Fry reviewed drafts of the paper, laboratory support.

## DNA Deposition

The following information was supplied regarding the deposition of DNA sequences:

Sequences are available on the European Nucleotide Archive (ENA) under project accession numbers PRJEB14267 and PRJEB20507.

## Data Availability

MOST, PneumoCaT and kmerID script code is available at https://github.com/phe-bioinformatics.

## Supplemental Information

Supplemental information for this article can be found online at http://dx.doi.org/10.7717/peerj.3571#supplemental-information.

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
