# Peer review of "Clinical streptococcal isolates, distinct from Streptococcus pneumoniae, but containing the β-glucosyltransferase tts gene and expressing serotype 37 capsular polysaccharide"

_PeerJ, doi:10.7717/peerj.3571_

## Round 0.1 · original submission · Minor Revisions

· Academic Editor

Minor Revisions

Dear Dr. Sheppard,

Please see the reviewers comments. I appreciate you addressing their remaining concerns and resubmitting the manuscript.

·

Basic reporting

The paper is well written and well referenced.

Experimental design

The experimentals are of very high order

Validity of the findings

The data is in agreement with the conclusions and discussion

Additional comments

The data presented is adequate though I was unable to reach the raw data PRJEB20507 at the ENA (Does this reflect my inability to get there ?). Clearly the paper focuses on showing that some clinical isolates non pneumococcal species of Streptococcus contain the tts gene while showing the presence of other genes. The paper shows that SNPS indicated greater similarity with S.mitis or S.pseudopneumoniae. I would suggest that the paper may be improved by placing the discussion of the tts proteins sequence in the section "Genomic analysis results using PHE S.pneumoniae workflow" rather than a separate section "Analysis of the tts sequences". Since that is an important part of the paper the nucleotide and protein sequence would better be discussed together. It is not clear as to whether the differences between the amino acid sequence alters the structure of the protein and hence influences its activity.. Part of the sequence tts gene is amazingly conserved across diverse bacteria and hence may provide some survival advantage Similarly all the genomic data and may be clubbed together. The sections Further genomic analysis and Genomic analysis results using PHE S.pneumoniae workflow may be integrated.
One issue that could have been raised in the paper is regarding documented data that non capsular streptococci actually appear to pick up external genes through horizontal transfer (Chrispin Chaguza, Jennifer E. Cornick, and Dean B. Everett. 2015 Mechanisms and impact of genetic recombination in the evolution of Streptococcus pneumonia Comput Struct Biotechnol J. 13: 241–247) to a larger extent than do the capsular types. Moreover even capsular Streptococci often use the non capsulated from for invasion of the upper respiratory tract (Carvalho et al. 2013, Non-pneumococcal mitis-group streptococci confound detection of pneumococcal capsular serotype-specific loci in upper respiratory tract. PeerJ 1:e97; DOI 10.7717/peerj.97) . The genomic data could have been used either to refute or buttress the issue. This leads to whether current taxonomy of the Streptococci is relevant today?
Does the use of vaccines increase the availability or discovery of non capsular Streptococci in the respiratory tract ?

Reviewer 2 ·

Basic reporting

No comment

Experimental design

No comment

Validity of the findings

No comment

Additional comments

Authors describe presence of tts gene and capsule production of serotype 37 in non-pneumococcal isolates. With the increased use of WGS for serotyping and genotyping of pneumococci there are increased reports in the literature of pneumococcal serotypes being described in streptococci other than S.pneumoniae. This paper add to this increasing body of information. The paper is well written and I don't have any additional comments or edits other than query as to whether these isolates were also tested with Quellung in addition to slide agglutination test.

---

## Round 0.2 · accepted · Accept

· Academic Editor

Accept

Thank you for addressing the concerns raised by the reviewers and systematically revising your manuscript. Congratulations! I am happy to recommend your article for publication in PeerJ.